# A Systematic Review of Naldemedine and Naloxegol for the Treatment of Opioid-Induced Constipation in Cancer Patients

**DOI:** 10.3390/pharmacy12020048

**Published:** 2024-03-06

**Authors:** Ursula K. Braun, Leanne K. Jackson, Mary A. Garcia, Syed N. Imam

**Affiliations:** 1Rehabilitation & Extended Care Line, Michael E. DeBakey VA Medical Center, Houston, TX 77030, USA; leanne.jackson2@va.gov (L.K.J.); maryacelle.garcia@va.gov (M.A.G.); 2Section of Geriatrics & Palliative Medicine, Baylor College of Medicine, Houston, TX 77030, USA; syed.imam@va.gov; 3Office of Connected Care, Michael E. DeBakey VA Medical Center, Houston, TX 77030, USA

**Keywords:** naldemedine, naloxegol, opioid-induced constipation, obstipation, cancer patients

## Abstract

Background: Opioid-induced constipation (OIC) is a pervasive and distressing side effect of chronic opioid therapy in patients with cancer pain, significantly impacting their quality of life. Peripherally acting μ-opioid receptor antagonists (PAMORAS) were developed for treatment-resistant OIC but most studies were conducted with non-cancer patients. Objective: to discuss two oral formulations of PAMORAs, naldemedine and naloxegol, and to review available evidence of the effectiveness of these drugs for OIC in cancer patients. Methods: a comprehensive search to identify primary literature for either naldemedine or naloxegol for OIC in cancer patients. Results: Only three prospective randomized, double-blind, placebo-controlled clinical trials for naldemedine enrolling cancer patients were identified; the results of a subgroup analysis of two of those studies and two non-interventional post marketing surveillance studies of these trials are also reported here. For naloxegol, only two randomized controlled trials were identified; both were unsuccessful in enrolling sufficient patients. An additional four prospective non-interventional observational studies with naloxegol were found that enrolled cancer patients. There were significantly higher rates of responders in the PAMORA groups than in the placebo groups. The most common side effect for both PAMORAs was diarrhea. Limitations: All studies were industry-funded, and given that only three trials were randomized controlled studies, the overall quality of the studies was lacking. Conclusion: Naldemedine or naloxegol appeared safe and useful in the treatment of OIC in cancer patients and may improve their quality of life. Larger-scale randomized placebo-controlled studies of PAMORAs in cancer patients would strengthen existing evidence.

## 1. Introduction

Opioid-induced constipation (OIC) is a disabling symptom which 60–90 percent of cancer patients with chronic opioid use experience [1,2,3,4]. Opioids bring about analgesia largely by binding to μ-receptors in the central nervous system, but they also bind to the μ-receptor in the myenteric plexus in the gastrointestinal tract, leading to the adverse side effect of constipation by decreasing intestinal motility, increasing fluid and electrolyte absorption in the small intestine and the colon, while also increasing the anal sphincter tone [1,2,3,4]. This can lead to more water absorption from the feces resulting in hard and dry stool. OIC has been defined by the Rome IV criteria as worsening symptoms of constipation when initiating, changing, or increasing opioid therapy, and it must include at least two of the following: fewer than three spontaneous bowel movements per week, straining during more than one-fourth of defecations, lumpy or hard stools in more than one-fourth of defecations, sensation of incomplete evacuation in more than one-fourth of defecations, or manual maneuvers to facilitate more than one-fourth of defecations (e.g., digital evacuation, support of the pelvic floor) [5,6].

Peripherally acting μ-opioid receptor antagonists (PAMORAs) are a class of medications aiming to reverse opioids’ adverse effects on the gut by interacting with opioid receptors in the gastrointestinal tract without significantly crossing the blood–brain barrier, and therefore they are not affecting the analgesic opioid effects in the central nervous system [7,8,9,10,11,12]. They are different from classic laxatives as, by their mechanism, they are targeted therapies for OIC. PAMORAs have been approved in the US by the Federal Drug Administration for OIC in patients with chronic non-cancer pain [13,14], and in Europe by the European Medicines Agency for use in patients with or without cancer [15,16]. In the US, naloxegol [12.5, 25 mg] was approved in September 2014, and naldemedine [0.1, 0.2 mg] was approved in March 2017 [13,14]. Patients with OIC can suffer greatly from reduced quality of life, as some may reduce their opioid dose in attempts to ease the OIC, leading to inadequate analgesia and a vicious circle without adequate relief of OIC. The American Gastroenterological Association published guidelines for the management of OIC [17], and other societies have published guidelines for the management of constipation in patients with cancer which specifically target OIC by including PAMORAs [18,19].

The objective of this review was to describe available primary literature on the use of oral naldemedine (sold as Symproic^®^ in the US or Rizmoic^®^ in the European Union) and oral naloxegol (sold as Movantik^®^ or Moventig^®^), specifically in cancer patients.

### 1.1. Mechanism of Action

PAMORAs are used in the treatment of opioid-induced constipation because they block and competitively prevent the binding of opioid agonists to μ-opioid receptors in the gastrointestinal tract [7]. PAMORAs act on gut motility, gut secretion and sphincter function [8]. Opioid agonists induce decreased cyclic adenosine monophosphate (cAMP) formation, and this effect is reversed by PAMORAs, leading to normalized chloride secretion. PAMORAs’ effect on gut motility leads to decreased transit time. This reduces the passive absorption of water from the stool, thus allowing for less dry and hard stools [9]. PAMORAs can also prevent sphincter of Oddi dysfunction and anal sphincter dysfunction caused by opioids, reducing straining and incomplete emptying.

### 1.2. Structure

Naloxegol and naldemedine are structurally similar to morphine and other μ-opioid receptor agonists. They both have a pentacyclic structure with a benzene ring, tetrahydrofuran ring, two cyclohexane rings, and a piperidine ring. The phenolic ring and its 3-hydroxyl group play a central role in the analgesic effects of opioids, as removal of the OH group reduces analgesic activity significantly. Naldemedine, with a chemical formula of C_32_H_34_N_4_O_6_, is a peripherally acting μ-opioid receptor antagonist derived from naltrexone. It blocks opioid receptors of the μ, δ, and κ types in the gastrointestinal tract. Patents for naldemedine tosylate are expected to expire between 2026 and 2031. Unlike naltrexone, which can cross the blood–brain barrier and is used to treat opioid dependence, naldemedine has a large hydrophilic side chain and affinity to P-glycoprotein, resulting in minimal concentrations in the central nervous system. Due to its low abuse potential, the Drug Enforcement Administration removed naldemedine from Class II scheduling in September 2017 [20]. Naloxegol oxalate (chemical formula C_34_H_53_NO_11_) is another peripherally acting μ-opioid receptor antagonist (PAMORA) and a PEGylated derivative of naloxol, a derivative of naloxone (chemical formula C_19_H_23_NO_4_) [21]. It also does not cross the blood–brain barrier and is not a Class II schedule drug [22].



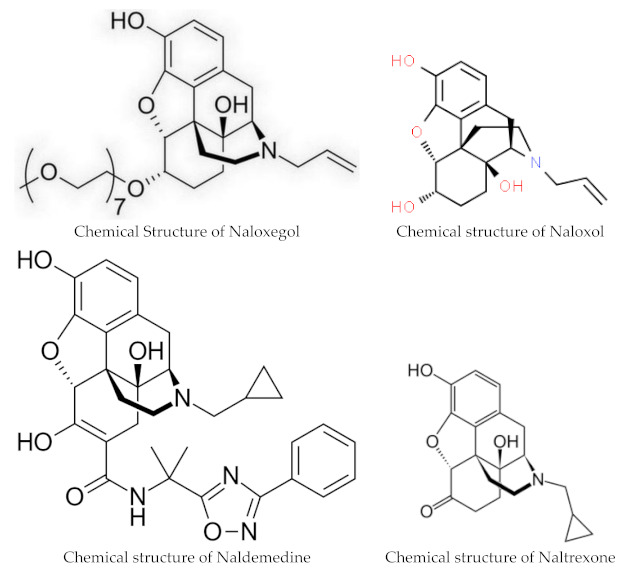



### 1.3. Pharmacokinetics

The oral bioavailability of Naldemedine ranges from 20% to 56%, with peak blood plasma levels achieved after 45 min on an empty stomach and 150 min when taken with a high-fat meal. The substance is highly bound to plasma proteins, primarily albumin, in the blood. The recommended dosage is 0.2 mg once daily with or without food [13,15]. Naldemedine is primarily metabolized by CYP3A to nor-naldemedine, and to a lesser extent by UDP-glucuronosyltransferase 1A3 to naldemedine 3-Glucuronide. Both metabolites are opioid receptor antagonists, but they are less potent than the original drug [23]. The drug is excreted in urine and feces, with an elimination half-life of 11 h. Patients with severe hepatic impairment should avoid naloxegol or naldemedine, although both drugs have been found to be safe and effective for those with mild to moderate hepatic impairments [24,25].

Naldemidine requires no adjustment for renal impairment [13,24,26]. For naloxegol, it is recommended that patients with a creatinine clearance <60 mL/min start with the lower naloxegol dose of 12.5 mg once daily and then, if tolerated, can increase the dose to 25 mg once daily [14,27].

Naloxegol clears mostly via hepatic metabolism (P450-CYP3A) with unknown actions of the metabolites. Naloxegol is excreted mostly in feces (and to some degree in urine), and its elimination half-life is 6–11 h [14]. Like naldemedine, when naloxegol is given with a fatty meal, absorption increases. Naloxegol is given as a once daily 12.5 or 25 mg tablet daily, and it should be taken on an empty stomach 1 h before or 2 h after the first meal of the day. Naloxegol may be crushed and can be given via nasogastric tube [14]. Maintenance laxatives should be discontinued prior to starting PAMORA therapy but may be resumed if OIC persists after 3 days of daily treatment. Decreased need for other laxatives may significantly reduce pill burden as, in many studies, PAMORAs alone were sufficient for relief of OIC.

### 1.4. Interactions

Even though PAMORAs in the US were approved for treatment of OIC in adults with noncancer pain, they have also been approved in other countries for both cancer and non-cancer pain [15,16], and are often prescribed off-label for OIC in cancer patients in the US. A very low risk of opioid withdrawal exists, so patients starting PAMORAs should be monitored for withdrawal symptoms such as hyperhidrosis, rhinorrhea, anxiety, and chills, though this was not commonly observed in clinical real-world practice. Opioid antagonists such as naloxone and naltrexone should not be used in conjunction with PAMORAs because of the potential increased risk of withdrawal. Both Naldemidine and naloxegol are no longer considered Schedule II controlled substances [20,22].

Naldemedine undergoes primary metabolism by the liver enzyme CYP3A4. Inhibitors of this enzyme can elevate naldemedine levels in the body, potentially leading to more side effects (see Table 1). Drugs like itraconazole, ketoconazole, clarithromycin, and grapefruit juice are examples of such inhibitors. In contrast, substances such as rifampicin and St John’s wort, which induce CYP3A4 activity, can significantly decrease naldemedine concentrations.

### 1.5. Contraindications

Both naloxego and naldemidine are contraindicated in patients with gastrointestinal obstruction or patients with hypersensitivity to the medication. Naloxegol should be avoided with strong CYP3A4 inhibitors like clarithromycin and ketoconazole, as they can raise Naloxegol levels and increase the risk of side effects. If taking moderate CYP3A4 inhibitors, such as diltiazem, erythromycin, or verapamil, the dosage of Naloxegol needs to be reduced. Grapefruit and grapefruit juice may also increase Naloxegol levels. Rifampin, a CYP3A4 inducer, may reduce the effectiveness of Naloxegol.

### 1.6. Side Effects

The most common side effects of PAMORAs are diarrhea, abdominal pain, nausea, flatulence, vomiting and headache [7]. As pure opioid antagonists, Naloxegol and Naldemedine have no potential for abuse. During subgroup analysis of the COMPOSE trials I–III, no increase in adverse events (45.9%) for patients aged ≥65 years (*N* = 344) were found for naldemedine 0.2 mg compared to the overall group (47.1%) or compared to the placebo (51.6%), nor was there a difference in proportion of responders between older adults compared to the overall group [28]. Other subgroup analyses also found no increase in adverse events for naldemedine users with renal impairments [26] or in patients with hepatobiliary impairments from pancreatic cancer [29]. Moderate and strong CYP3A4 inhibitors and P-glycoprotein inhibitors may increase naldemedine concentrations; therefore, monitoring for adverse reactions is recommended in patients taking these medications. PAMORAs can improve quality of life, are generally safe and well tolerated, and offer a good response without reducing opioid-mediated analgesia. 

### 1.7. Clinical Trials

Naldemedine was approved based on the results of the Japanese-led COMPOSE trials, which were phase three clinical studies in adult outpatients with chronic non-cancer pain and opioid-induced constipation. COMPOSE-I and COMPOSE-II were 12-week double-blind multi-country randomized controlled trials comparing 0.2 mg oral once daily naldemedine with a placebo between 2013–2015 [30]. Responders had to have at least three spontaneous bowel movements per week, with an increase of one spontaneous bowel movement for nine of the twelve weeks; the proportion of responders were significantly higher in the naldemedine group in both trials. COMPOSE-III tested the long term safety of naldemedine in patients with non-cancer chronic pain over 52 weeks, finding a statistically significant increase in weekly bowel movements without any evidence of opioid withdrawal symptoms [31].

While there is ample literature on the use of PAMORAs in patients with non-cancer pain, recruiting seriously ill patients with cancer into clinical research trials outside of cancer-directed treatment trials is difficult due to the patients’ short life expectancy, impaired functional status, and high symptom burden [32,33].

The objective of this systematic review was to search for high quality prospective interventional trials of either naldemedine or naloxegol for OIC in *patients with cancer*.

## 2. Materials and Methods

### 2.1. Focal Question

This (unregistered) literature search identified all primary literature (original research articles, reports, conference papers) using either naldemedine or naloxegol for opioid-induced constipation in cancer patients.

### 2.2. Search and Information Sources

The MESH search terms ‘naldemedine’, ‘naloxegol’, ‘constipation’, ‘obstipation’, ‘opioid-induced constipation’, and ‘cancer’ were used to search PubMed, Cochrane Library, Medline (Ovid), Scopus, Embase, ClinicalKey, and CINAHL Plus databases, as mentioned in the articles’ title, abstract, or body, from 2000 to 2023. The search terms were entered into each database using the Boolean operator ‘AND’ in several combinations: ‘naldemedine’ AND ‘constipation, ‘naldemedine’ AND ‘opioid-induced constipation’, and ‘naldemedine’ AND ‘obstipation’. The same search was completed using the word ‘naloxegol’ rather than ‘naldemidine’. Additional searches were conducted through Google Scholar and reviewing of references of the discovered studies. The literature search was conducted between October and early December of 2023.

### 2.3. Eligibility Criteria and Study Selection

There was no restriction on article types screened, but only prospective clinical trials using naldemidine or naloxegol for the treatment of OIC were eligible for review. There also was no restriction on language used, though our search was conducted in English and would likely not have found articles in other languages if they did not have at least an abstract written in English. Each article’s title and abstract were reviewed for relevance, and only studies conducted with cancer patients were included. 

### 2.4. Outcomes Assessed

All studies used the ROME-IV criteria to define opioid-induced constipation, i.e., participants had to experience new or worsening constipation symptoms when starting or increasing opioid therapy and were straining during more than 25% of time spent passing stools, or had hard or lumpy stools more than 25% of the time [5,6]. Studies assessed changes in spontaneous bowel movement (SBM) frequency/week from baseline, SBM with complete bowel evacuation/week, and SBMs without significant straining/week for both intervention and placebo groups. Some studies also looked at the median time to first SBM after drug administration. Quality of life outcomes were assessed using the PAC–SYM stool domain and PAC–QOL dissatisfaction domain. All studies recorded adverse events and the seriousness of adverse events.

### 2.5. Quality Assessment

Due to the paucity of studies meeting all our inclusion criteria (being prospective controlled studies of either naldemedine or naloxegol enrolling cancer patients with OIC), we also report results of prospective post-marketing surveillance extension studies, which obviously were of much lower quality due to lack of a comparison group. We used the Critical Appraisal Skills Programme (CASP) [34] to assess the studies, which is appropriate for randomized controlled trials. 

## 3. Results

Figure 1 shows a flowchart describing how the articles were chosen for inclusion in this review.

### Retrospective Studies Excluded

We excluded retrospective studies; many were conducted in patients with non-cancer, but several were conducted in special populations of cancer patients. One multi-center study evaluated the efficacy and safety of naldemedine in 40 hospitalized thoracic cancer patients with opioid-induced constipation in Japan where 65% of patients were responders [35]. Diarrhea was the most common adverse event and occurred in 27.5% (11 patients) but was mild for most of them (9 patients). Another Japanese multi-center retrospective study examined 33 hospitalized patients with gastrointestinal cancers (esophageal, gastric, small bowel, and colorectal cancers) of which, again, 63.6% were considered responders, i.e., they had a significant increase in bowel movement frequency of at least three times per week or at least once per week above the baseline after initiation of naldemedine [36]. Mild diarrhea was again the most common adverse event and occurred in 39.45% (13 patients). Additionally, 34 hospitalized patients with hepatobiliary pancreatic cancer (liver, biliary tract, and pancreatic cancers) who were taking opioids and received naldemedine during hospitalization were the focus of another retrospective multicenter Japanese study and were assessed for frequency of bowel movements before and after starting naldemedine [29]. In this group, 21 were responders (61.7%), defined as having ≥3 bowel movements/week, or with an increase from a baseline of ≥1 bowel movement/week over seven days after beginning daily naldemedine administration. The median number of weekly bowel movements before and after naldemedine treatment was two (range: 0–9) and six (range: 1–17), respectively; the increase in the number of bowel movements following naldemedine administration was statistically significant (Wilcoxon signed-rank test, *p* < 0.0001). Almost one third of patients experienced diarrhea as the most common side effect. 

Cancer patients with a poor performance status are a particularly vulnerable group whose quality of life can be significantly impaired by constipation. Another retrospective multi-center Japanese study evaluated 71 cancer patients with ECOG performance status, 3 or 4 of whom received naldemedine [37]. Of these, 66.1% responded, i.e., bowel movement frequency increased to ≥3 times/week over 7 days after naldemedine administration., and grade 1 or 2 diarrhea was again the most common adverse event (38%). All of these retrospective studies in different types of cancer patients showed that naldemedine seemed safe and effective in clinical real-world practice, regardless of type of cancer and the performance status of patients.

We used the Critical Appraisal Skills Programme (CASP) [34] to assess the studies we included in our review, which is an appropriate tool for randomized controlled trials. All of the studies we found were industry-sponsored, with authors having conflicts of interest, making them prone to bias. Only five of the studies were randomized blinded placebo-controlled studies [32,33,38,39,40], two of which were unable to enroll sufficient participants [32,33]. The other seven studies were all non-interventional prospective open label extension/post-marketing surveillance studies [41,42,43,44,45,46,47], and therefore the quality of most of the discussed studies was not very high.

Table 2 describes the three randomized placebo-controlled trials included in this review for naldemedine [38,39,40], as well as results from a pooled subgroup analysis of patients from both 2017 Katakami studies [14] and two post-marketing surveillance studies of naldemidine [42,43]. It also details the six studies included for nalexogol, of which two were unsuccessful randomized controlled clinical studies due to insufficient enrollment [32,33], and four were non-interventional prospective observational studies in cancer patients lasting between 4 weeks and 1 year [44,45,46,47]. Studies described safety and efficacy, and the naloxegol studies included outcomes on quality of life. Table 2 describes the study design, medication regimen, inclusion criteria, outcomes assessed, and the results of each trial in detail.

The most common side effects described with both nalexogol and naldemedine were diarrhea, abdominal pain, abdominal bloating, nausea, and dysesthesia; almost all adverse effects were of a mild degree.

## 4. Discussion

Our literature search revealed three randomized placebo controlled double-blind trials in patients with cancer using naldemedine to treat OIC, as well as two unsuccessful randomized placebo-controlled double-blind trials using naloxegol for OIC in patients with advanced cancer and cancer-related pain that were closed early due to poor patient enrollment. Additionally, three non-interventional studies of naldemidine and four non-interventional studies of naloxegol in cancer patients were found. Large interventional studies with cancer patients were absent, reflecting the enormous recruiting difficulties in this population; thus, higher quality studies are only available for naldemidine but not for naloxegol. All naldemidine studies were conducted in Japan, whereas all naloxegol studies were non-interventional and conducted in Europe. Attempts at high-quality randomized controlled trials with naloxegol failed due to enormous recruiting difficulties in the US. 

The usual medication doses studied were 0.2 mg for naldemedine and 12.5 or 25 mg for naloxegol, and, at these doses, treatment emergent adverse events were usually relatively minor. Cancer patients with OIC seem to be ideal candidates for treatment with PAMORAs as they block the constipating effects of opioids at the myenteric plexus without affecting analgesia in the central nervous system. Ongoing treatment for the disabling opioid side effect of constipation with PAMORAs seems to be safe, even if given for prolonged periods of times of up to one year based on the open label extension studies. No particular monitoring for toxicity is required, and even patients with renal or hepatic impairment up to CHILD class B are good candidates for PAMORAs without a need for dose adjustment [24,25,26,27].

While conducting studies on symptom management in cancer patients seems difficult and fraught with recruitment challenges, head-to-head comparisons of laxatives versus PAMORAs or different PAMORASs against each other are even rarer, even in patients with non-cancer pain who are easier to recruit. We only found one retrospective study comparing oral naloxegol with subcutaneous methylnaltrexone [48]; however, this study was conducted in seriously ill patients in an intensive care unit setting and included only 30 cancer patients (15% of patients in the naloxegol group and 28% in the naltrexone group had cancer). They found that both drugs increased SBMs and that naloxegol was non-inferior to subcutaneous methylnaltrexone at a significantly reduced cost. 

### 4.1. Limitations and Study Quality

We reviewed all prospective studies using either naldemidine or naloxegol in patients with OIC and cancer; all these studies were funded by industry, and investigators were either employees of the drug companies or received royalties from the drug companies sponsoring the studies, making them prone to bias. However, the three RCTs by Katakami [38,39,40] were conducted according to generally accepted standards with appropriate randomization and blinding, following a well-described study protocol and reporting results comprehensively, including confidence intervals and reporting of adverse events (these RCTs were also comparable since the outcome measures used were similar for these trials). Of the 12 studies reviewed, only 5 were RCTs, and of those only 3 (naldemidine studies) [38,39,40] had results, as the other 2 (naloxegol studies) [32,33] were unable to recruit enough patients. The other studies reviewed were various non-interventional open label extension studies of the primary Katakami studies (for naldemidine) or non-interventional observational studies (for naloxegol) [44,45,46,47], and thus of much lower quality. However, we reviewed all in the absence of any better evidence, as our clinical question was about use of PAMORAs specificially in cancer patients with OIC. The ‘real world’ studies were also the only studies able to follow patients taking PAMORAs over a longer period of time, While the RCTs only administered the study drug for two weeks [33,38,39,40], the ‘real world’ open label extension studies were either up to four weeks [45,46], three months [39,40,41,42,43,44] or even up to one year [47] long.

Most studies were conducted in Japan [38,39,40,41,42,43], and some in Europe [44,45,46,47], while the few studies conducted in the US [32,33] were unable to recruit enough participants; therefore, it is not clear if results would be fully applicable to patients in North America. 

### 4.2. Implications for Pharmacy Practice

Pharmacists need to take an active role in counseling patients who are taking opioid medications for cancer pain and who will invariably experience OIC. OIC significantly impairs the quality of life of patients requiring opioids for pain control, which makes prevention, identification, and treatment of OIC, as well as counseling on available medications and their proper use, critical areas for pharmacist involvement. It is important to note that the usual first line counseling techniques by clinical pharmacists, like increasing hydration and physical activity, even if necessary in a global cancer context, is not effective in OIC. Similarly, traditional laxatives may not be effective in OIC but can be responsible for side effects, enormous pill burden, and, in some cases, patients reducing their opioids, which then leads to increased suffering due to inadequate pain control. Therefore, being knowledgable about PAMORAs to treat OIC is crucial for clinical pharmacists who assist cancer patients. Due to their increased cost, large health care systems like the Veteran Health Administration have developed criteria for the use of both Naldemedine [49] and naloxegol [50] t by theVA Pharmacy Benefits Management Services.

## 5. Conclusions

Naldemedine at 0.2 mg or naloxegol at 12–25 mg once daily were useful in the treatment of OIC in cancer patients, improving their quality of life while producing relatively minor side effects. Both drugs have high potential in palliative and hospice care due to the debilitating effects of treatment-resistant OIC and non-responsiveness to traditional laxatives. Clinical pharmacists take an active role in counseling patients on the appropriate use of PAMORAs, which can have a tremendous impact on cancer patients’ well-being. Few high quality trials were found for naldemidine and none for naloxegol, and the trials for the former were of short duration (14 days). Further longer-lasting larger-scale randomized placebo-controlled or comparative studies with standard laxative medications and PAMORAs in cancer patients would strengthen existing evidence but are wrought with significant recruitment challenges.

## Figures and Tables

**Figure 1 pharmacy-12-00048-f001:**
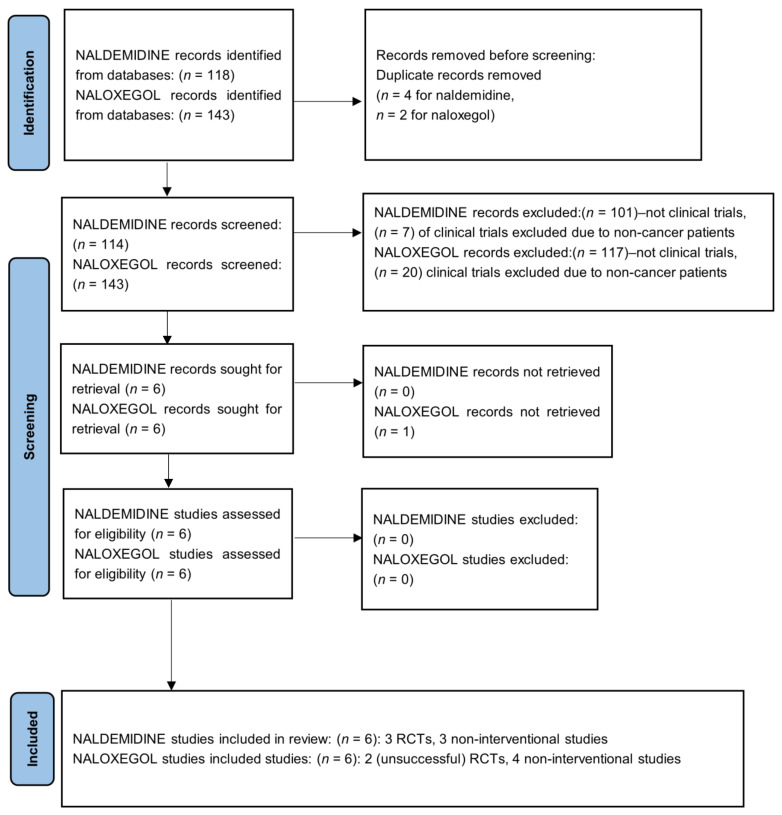
Description of identification of studies included in this review (Von Roenn et al., 2013) [32].

**Table 1 pharmacy-12-00048-t001:** Inhibitors and inducers of CYP3A4 which potentially increase or decrease naldemedine concentrations.

Strong Inhibitors of CYP3A4: Increase Naldemedine Concentration	Inducers of CYP3A4: Decrease Naldemedine Concentration
Itraconazole	Rifampine
Ketoconaxole	St. John’s wort
Clarithromycin	
Grapefruit juice	
**Moderate Inhibitors of CYP3A4:** Diltiazem Erythromycin Verapamil	

Potent inhibitors of the P-glycoprotein pump, like ciclosporin, have the potential to elevate naldemedine blood levels.

**Table 2 pharmacy-12-00048-t002:** Primary literature related to naldemedine or naloxegol for opioid-induced constipation in cancer patients.

Authors	Study Design	Medication Regimen	Inclusion Criteria	Outcomes Assessed	Results
Katakami N et al., J Clin Onc, June 2017, Japan [38]	Phase II b randomized double-blind placebo-controlled study	Naldemedine 1:1:1:1:1 assigned to either 0.1 mg, 0.2 mg, 0.4 mg, or placebo oral daily for 14 days	Adults 18 years or older with OIC and cancer, ECOG ≤ 2, on stable opioid regimen for ≥2 weeks	Primary: Change in spontaneous bowel movement frequency/week from baseline Secondary: - SBM responder rates - Change from baseline in frequency of SBM without straining - Complete SBM - Safety	- *N* = 227 (55–58/group) - Change in SBM frequency higher with all naldemedine doses vs. placebo (*p* < 0.05), as were SBM responder rates and change in complete SBM frequency - Change in SBM frequency without straining significantly improved with naldemedine 0.2 and 0.4 (but not 0.1) mg vs. placebo (at least *p* < 0.05).
					- Adverse events were more common with naldemedine (0.1 mg: 66.1%; 0.2 mg: 67.2%; 0.4 mg: 78.6%) than placebo (51.8%) - Most common adverse event: diarrhea
Katakami N et al., J Clin Onc, December 2017, Japan [39]	COMPOSE-4: randomized Phase III placebo-controlled double-blind study COMPOSE-5: open-label extension study	COMPOSE-4: 1:1 random assignment to Naldemedine 0.2 mg vs. placebo daily for 14 days COMPOSE-5: open-label 12-week extension	Adults 20 years or older with OIC and cancer, ECOG ≤ 2, on stable opioid regimen for ≥2 weeks	COMPOSE-4 Primary endpoint: Proportion of SBM responders (≥3 SBMs/week and increase ≥1 SBM/week from baseline) COMPOSE-5 primary end point: safety.	COMPOSE-4: *N* = 193 (97 naldemedine, 96 placebo); COMPOSE-5: *N* = 131 - Proportion of SBM responders naldemedine vs. placebo (71.1% [69 of 97 patients] vs 34.4% [33 of 96 patients]; *p* < 0.0001). - Greater change from baseline with naldemedine than with placebo in frequency of SBMs/week (5.16 v 1.54; *p* < 0.0001), SBMs with complete bowel evacuation/week (2.76 v 0.71; *p* < 0.0001), and SBMs without straining/week (3.85 v 1.17; *p* = 0.0005).
					- Treatment-emergent AEs: higher in patients treated with naldemedine than with placebo (44.3% [43 of 97 patients] v 26.0% [25 of 96 patients]; *p* = 0.01) - in COMPOSE-5, 105 (80.2%) of 131 of patients reported TEAEs. - Diarrhea was the most frequently reported TEAE in COMPOSE-4 (19.6% [19 of 97 patients on naldemedine] vs. 7.3% [seven of 96 patients] on placebo) and COMPOSE-5 (18.3% [24 of 131 patients] with naldemedine). - Naldemedine was not associated with opioid withdrawal and had no notable impact on opioid-mediated analgesia.
Katakami N et al., Ann Onc, 2018, Japan [40]	COMPOSE-4: randomized Phase III placebo-controlled double-blind study COMPOSE-5: open-label extension study	COMPOSE-4: 1:1 random assignment to Naldemedine 0.2 mg vs. placebo daily for 14 days COMPOSE-5: open-label 12-week extension study of Naldemedine 0.2 mg	Adults 20 years or older with OIC and cancer, ECOG ≤ 2, on stable opioid regimen for ≥2 weeks	Secondary endpoints: - proportion of complete SBM (CSBM) responders, SBM or CSBM responders by week - subjects with ≥1 SBM or CSBM within 24 h post-initial dose - Changes from baseline in frequency of SBMs or CSBMs/week assessed at weeks 1 and 2 - Time to first SBM or CSBM postinitial dose - QOL impact was evaluated by Patient Assessment of Constipation-Symptoms (PAC-SYM) and PAC-QOL questionnaires	- *N* = 193 for COMPOSE-4, *N* = 1341 for COMPOSE-5 - Improved bowel function for all secondary efficacy assessments versus placebo (all p ≤ 0.0002). - Median time to first SBM (4.7 h versus 26.6 h) and CSBM (24.0 h versus 218.5 h) post initial dose (all *p* < 0.0001). In COMPOSE-4, significant differences between groups were observed with the PAC-SYM stool domain (*p* = 0.045) and PAC-QOL dissatisfaction domain (*p* = 0.015). In COMPOSE-5, significant improvements from baseline were observed for overall and individual domain scores of PAC-SYM and PAC-QOL.
					- No discussion of AE/SE other than mentioning that naldemidine did not lead to any signs/symptoms of opioid withdrawal
Osaka I et al., Esmo Open 2019, Japan [41]	Subgroup analysis of pooled data from both Katakami 2017 studies	Naldemedine 0.2 mg vs. placebo	Adults 18 years or older with OIC and cancer, ECOG ≤ 2, on stable opioid regimen for ≥2 weeks	Proportions of SBM responders and patients with diarrhea. For patient subgroups with or without possible blood–brain barrier (BBB) disruptions, changes in Numerical Rating Scale (NRS) and Clinical Opioid Withdrawal Scale (COWS) scores.	- *N* = 307 (naldemedine: *n* = 155; placebo: *n* = 152) - 73.5% SBM responders in naldemedine group versus 35.5% with placebo. - Significant increase in the proportion of SBM responders with naldemedine versus placebo (38.0% (95% CI 27.6% to 48.4%); *p* < 0.0001).
					- Changes from baseline in NRS and COWS scores were similar with naldemedine or placebo in patients with/without brain metastases - Higher proportions of SBM responders and patients who experienced diarrhea were observed with naldemedine versus placebo in all subgroups.
Takata K et al., Support Care Cancer 2022, Japan [42]	Non-interventional multi-center prospective post-marketing surveillance	Naldemedine 0.2 mg, for up to 12 weeks	Adult patients with opioid-induced constipation (OIC) and cancer pain	Safety & effectiveness	Effectiveness analysis set (*N* = 953): Improved frequency (75.0% and 83.2%) and condition of bowel movement (80.0% and 88.0%) at 2 and 12 weeks, respectively
					Safety analysis set (*N* = 1177), 145 ADRs occurred in 133 (11.3%) patients, diarrhea was the most frequent event (*n* = 107, 9.09%) but most cases of diarrhea were non-serious (98.1%). Most ADRs were non-serious (93.8%) and resolved within 2 weeks (75.9%).
Naya N, 2023, Cureus, Japan [43]	Non-interventional exploratory post hoc subgroup analysis of post-marketing surveillance, same dataset as [42]	naldemedine 0.2 mg, for up to 12 weeks	Adult patients with opioid-induced constipation (OIC) and cancer pain	Safety & effectiveness with subgroup analysis by: - age (≥75, <75 years) - ECOG performance status (PS 0–2, 3–4) - constipation severity (mild, moderate, severe) - brain metastasis (yes, no) - anticancer drug treatment (yes, no) - opioid at naldemedine initiation (fentanyl only, only strong opioids other than fentanyl, weak opioids only, other), - prior or concomitant use of laxative (only osmotic/saline laxatives, only stimulant laxatives, other, none)	- *N* = 1184 - Through week 2 to week 12, improvement rates in the frequency and condition of bowel movement among subgroups ranged from 63.6% to 89.7% and 67.6% to 94.9%, compared to 75.0% to 83.2% and 80.0% to 88.0% in the total population, respectively.
					Incidence of AE, including diarrhea, among subgroups ranged from 7.74% to 16.08% (diarrhea: 5.95% to 13.19%), compared to 11.30% (diarrhoea: 9.09%) in the total population.
Von Roenn JH et al., 2013, USA, published as poster only [32]	KODIAC-06, planned as a randomized, placebo-controlled, double-blind, multicenter, phase 3 trial	Naloxegol 12.5 or 25 mg	Adult cancer patients with OIC	- Efficacy	Study was closed early due to inability to enroll sufficient patients. No further details available, no response from author received by date of submission.
Bull J et al., J Pain Sym Man, 2019, USA [33]	Feasibility study, planned as 3-center randomized, placebo-controlled trial	Naloxegol 25 mg, with or without concomitant use of laxatives, (14 days of double-blind naloxegol vs. placebo followed by 14-day open-label naloxegol daily)	Adult advanced cancer patients aged ≥ 18 years, with life expectancy > 8 weeks, PPS ≥ 30, on at least 20 Morphine equivalents/d for >1 week, with OIC on laxatives	- Feasibility of a definitive trial for OIC in advanced cancer patients - Secondary outcomes: tolerability, safety, and efficacy.	Study closed early after 24 months due to inability to enroll sufficient patients: - An amount of 590 screened, 414 excluded for medical ineligibility, 140 patients/family declined, 24 other reasons ≥ only 12 patients participated. Of those, 7 became ineligible during the baseline/OIC confirmation period. Of the 5 randomized patients, 1 withdrew consent. Only 4 patients completed the study. No adverse events were reported related to the study drug.
Cobo Dols M et al., BMJ Support Palliat Care, 2020, Spain [44]	Non-interventional, 3-month follow-up observational cohortstudy	Naloxegol 12.5 or 25 mg, with or without concomitant use of laxatives	Adult cancer patients ≥ 18 years, on opioids for pain with OIC on laxatives, Karnofsky ≥ 50	- efficacy - quality of life	- *N* = 126 patients with mean age of 61.3 years, 6 patients withdrew (within first 30 days) - Number of days/week with complete SBMs increased significantly (*p* < 0.0001) from 2.4 to 4.6 on day 15, 4.7 after 1 month and 5 after 3 months. - Clinically relevant improvements (>0.5 points) in the PAC-QOL and PAC-SYM questionnaires (*p* < 0.0001) from 15 days of treatment. - Pain control significantly improved (*p* < 0.0001) during follow-up. - 13.5% of patients (17/126) had some gastrointestinal AE, mostly of mild (62.5%) or moderate intensity (25%).
Lemaire A. et al., Supp Care Cancer, 2021, France [45]	Non-interventional “real life” outpatient multi-center, 4-week follow-up observational study	Naloxegol 12.5 or 25 mg, with or without concomitant use of laxatives	Adult cancer patients aged 18–70 years old with OIC on laxatives, any ECOG, on any opioid regimen	- Response rate to naloxegol at week 4 (primary criterion) - Evolution of quality of life using the Patient Assessment of Constipation Quality of Life (PAC-QOL) questionnaire - Safety	*N* = 124 cancer patients of which 79% had ECOG ≤ 2, metastatic stage, 80%. At inclusion, the median opioid dosage was 60 mg of oral morphine or equivalent. - At week 4, the response rate was 73.4% (95% CI [63.7–83.2%]), and 62.9% (95% CI [51.5–74.2%]) of patients had a clinically relevant change in quality of life (decrease in PAC-QOL score ≥ 0.5 point). - 8% of patients had adverse events related to naloxegol (7% with gastrointestinal events; one serious diarrhea).
Davies A et. al., Cancers, 2022, 26 European countries [46]	Non-interventional, prospective “real world” singles arm open label multi-national 4-wek study	Naloxegol 12.5 or 25 mg (−50 mg), with or without concomitant use of laxatives	Adult cancer patients ≥ 18 years old who had been on opioids for at least 4 weeks and had OIC, any ECOG, on any opioid regimen. Colorectal cancer pts were excluded	- Safety,-defined as incidence of adverse events leading to study discontinuation - efficacy, - QoL	- *N* = 170 received at least one dose of naloxegol (=safety population) - Of 76 patients who completed both 4 weeks of treatment and 28 days of diary, 55 patients (72.4%, 95% CI 62.3–82.4%) responded to naloxegol treatment (i.e., had ≥3 SBMs/week and an increase of ≥1 SBM over baseline) - The Patient Assessment of Constipation—QoL Questionnaire total score and all its subscales improved from baseline to 4 weeks of follow up - *N* = 20 (11.8%, 95%CI 6.9–16.6) discontinued the study due to AE, and, of them, 12 (7.1%, 95%CI 3.2–10.9%) discontinued due to naloxegol-related AE industry-sponsored
Cobo Dols M et al., BMJ Support Palliat Care, 2023, Spain [47]	Non-interventional, 1-year prospective observational “real-world” study (continuation of Cobo 2020 study)	Naloxegol 12.5 or 25 mg, with or without concomitant laxative use	Adult cancer patients ≥ 18 years, on opioids for pain with OIC on laxatives, Karnofsky ≥ 50	Long-term efficacy, quality of life (QOL) and safety of naloxegol. Assessed by the patient assessment of constipation QOL questionnaire (PAC-QOL), the PAC symptoms (PAC-SYM), the response rate at day 15, and months 1-3-6-12, and global QOL (EuroQoL-5D-5L)	- *N* = 126 patients, mean age 61.5 years. 53 patients died during the study from their cancer. - PAC-SYM and PAC-QOL total score and all their dimensions improved from baseline (*p* < 0.0001). - At 12 months, 77.8% of patients responded to naloxegol. - Global QOL was conserved from baseline. 6 patients withdrew from the study due to AE (abdominal pain), in the first 15 days (3) or the first 30 days (3). - 28 adverse reactions, mainly gastrointestinal were observed in 15.1% of patients (19/126), with 75% (21) classified as mild, 17.9% (5) as moderate and 7.1% (2) as severe; most adverse reactions (67.9%) appeared during the first 15 days of treatment.

OIC = opioid-induced constipation, SBM = spontaneous bowel movement, CBSM = complete spontaneous bowel movement, AE = adverse events, TEAE = treatment emergent adverse events, NRS = numerical rating scale, COWS = clinical opioid withdrawal scale, ECOG = Eastern Cooperative Oncology Group (assessment of functional status), PAC-SYM stool domain = patient assessment of constipation-symptoms, PAC-QOL dissatisfaction domain = patient assessment of constipation-quality of life, QoL = quality of life.

## Data Availability

Not applicable.

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
