# Peer review of "A Systematic Review of Naldemedine and Naloxegol for the Treatment of Opioid-Induced Constipation in Cancer Patients"

_pharmacy, 2024, doi:10.3390/pharmacy12020048_

Round 1

Reviewer 1 Report

Comments and Suggestions for Authors

I am sincerely grateful to the editors for entrusting me with the responsibility of reviewing the manuscript "A systematic review of Naldemedine and Naloxegol for the treatment of opioid-induced constipation in cancer patients." It is an honor to be chosen as a reviewer for Pharmacy (MDPI), and I appreciate the opportunity to contribute to the rigorous evaluation of this scholarly work. Your confidence in my expertise is truly appreciated, and I am committed to providing constructive feedback that will enhance the quality and impact of the manuscript.

While the manuscript displays promising potential, certain concerns need to be thoroughly addressed by the authors before the journal can consider it for publication. The forthcoming revised version of the manuscript is eagerly anticipated, and I look forward to witnessing the authors' thoughtful incorporation of the outlined concerns to enhance the overall quality and readiness for publication.

1. While the authors have labeled their manuscript as a "systematic review," it is crucial to note a deviation from the PRISMA guidelines. The absence of adherence to the PRISMA checklist is noticeable both within the manuscript's content and its various sections, including the abstract. A meticulous examination reveals a lack of alignment with the globally accepted standards for systematic reviews outlined in the PRISMA guidelines. It is imperative that the authors thoroughly revise the entire manuscript, ensuring strict adherence to the PRISMA checklist to meet the established criteria for systematic reviews recognized worldwide. This revision is vital to uphold the credibility and robustness of the research presented.

Dear authors, kindly consider revising the manuscript sections in accordance with the PRISMA checklist guidelines, which can be accessed through the following links. Your attention to aligning the content with these guidelines will greatly enhance the manuscript's adherence to established standards for systematic reviews.

http://www.prisma-statement.org/?AspxAutoDetectCookieSupport=1

2. In order to enhance clarity and facilitate a more structured understanding of the research methodology, I recommend organizing the "Materials and Methods" section of this manuscript into well-defined subsections. Utilizing subsection titles such as "Focal question," "Language," "Databases," "Study selection," "Data extraction," and "Quality assessment" will provide readers with a systematic and comprehensive insight into the methodology employed. Please ensure meticulous attention to detail when addressing each of these subsections, offering readers a clear and concise roadmap to follow the intricacies of your research process. This approach will not only improve the overall readability of the manuscript but also contribute to a more thorough comprehension of your study's methodology.

3. I recommend incorporating a quality assessment tool to evaluate the robustness of this systematic review. Choose from the various available tools and integrate it into your research to provide a comprehensive evaluation of the study's quality. In the "Materials and Methods" section, elucidate the methodology behind the selected quality assessment tool. Clearly outline the criteria, scoring system, and rationale behind its application, aiming to enhance transparency and credibility in the evaluation of your research findings. This inclusion will contribute to a more thorough understanding of the reliability and validity of your systematic review. Please integrate a table within the document to present a thorough quality assessment report, complemented by accompanying text for clarity and context.

4. The discussion section of your manuscript requires improvement in its description. I recommend a thorough revision, where each included study is individually addressed within this section. Provide a comprehensive analysis of the main biases associated with each study, and discuss the primary and secondary endpoints to enhance the depth of your interpretation. Please ensure that the discussion does not duplicate information already presented in the table, aiming for a nuanced exploration of each study's nuances and contributing to a more insightful and well-rounded understanding of your research findings.

5. In the "Authors" column of your table, kindly insert reference numbers corresponding to the reference list. This adjustment will ensure a more organized and systematic alignment between the authors' names and their respective references, facilitating easier cross-referencing for readers and enhancing the overall coherence of your presentation.

6. I request that you enhance and modify Figure 1 by incorporating the PRISMA chart, accessible at the provided link, to elucidate the search process employed in your manuscript. This adaptation will contribute to a clearer visualization of your study's systematic approach, aligning it with recognized standards and facilitating a more transparent representation of the search methodology for your readers.

7. Elevate the concluding section of your manuscript by paving the way for future research directions. Delve into the researchers' mindset, guiding them towards exploring novel studies that directly address the research findings, with an emphasis on benefiting future patients. Additionally, revisit your findings in greater depth, elucidating the clinical relevance of your results. By providing a thorough exploration of potential implications and applications, you can significantly enhance the impact and applicability of your research, contributing to the advancement of knowledge and its practical implications in the field.

8. I believe that the inclusion of a structured abstract may not align seamlessly with the standards set by MDPI. It would be beneficial to reconsider the abstract format to ensure it adheres more closely to the specific guidelines and expectations established by MDPI for a more harmonious fit within their publishing standards.

Thank you for your patience.

Yours sincerely,
The Reviewer

Author Response

We thank the reviewers for their time in reviewing our manuscript and attention to detail. We have revised our manuscript taking all their excellent points into consideration.  Comments and Suggestions for Authors

I am sincerely grateful to the editors for entrusting me with the responsibility of reviewing the manuscript "A systematic review of Naldemedine and Naloxegol for the treatment of opioid-induced constipation in cancer patients." It is an honor to be chosen as a reviewer for Pharmacy (MDPI), and I appreciate the opportunity to contribute to the rigorous evaluation of this scholarly work. Your confidence in my expertise is truly appreciated, and I am committed to providing constructive feedback that will enhance the quality and impact of the manuscript.

While the manuscript displays promising potential, certain concerns need to be thoroughly addressed by the authors before the journal can consider it for publication. The forthcoming revised version of the manuscript is eagerly anticipated, and I look forward to witnessing the authors' thoughtful incorporation of the outlined concerns to enhance the overall quality and readiness for publication.

1. While the authors have labeled their manuscript as a "systematic review," it is crucial to note a deviation from the PRISMA guidelines. The absence of adherence to the PRISMA checklist is noticeable both within the manuscript's content and its various sections, including the abstract. A meticulous examination reveals a lack of alignment with the globally accepted standards for systematic reviews outlined in the PRISMA guidelines. It is imperative that the authors thoroughly revise the entire manuscript, ensuring strict adherence to the PRISMA checklist to meet the established criteria for systematic reviews recognized worldwide. This revision is vital to uphold the credibility and robustness of the research presented.

Dear authors, kindly consider revising the manuscript sections in accordance with the PRISMA checklist guidelines, which can be accessed through the following links. Your attention to aligning the content with these guidelines will greatly enhance the manuscript's adherence to established standards for systematic reviews.

http://www.prisma-statement.org/?AspxAutoDetectCookieSupport=1 [nonfunctional]

While we followed a systematic review approach using the PRISMA checklist, this was not a registered search or attempt at a meta-analysis but a focused review of a practical clinical question (what is the evidence for use of naldemidine or naloxegol for OIC in cancer patients?).  We have added multiple sub-sections to improve the flow as outlined below.

  1. In order to enhance clarity and facilitate a more structured understanding of the research methodology, I recommend organizing the "Materials and Methods" section of this manuscript into well-defined subsections. Utilizing subsection titles such as "Focal question," "Language," "Databases," "Study selection," "Data extraction," and "Quality assessment" will provide readers with a systematic and comprehensive insight into the methodology employed. Please ensure meticulous attention to detail when addressing each of these subsections, offering readers a clear and concise roadmap to follow the intricacies of your research process. This approach will not only improve the overall readability of the manuscript but also contribute to a more thorough comprehension of your study's methodology.

We thank the reviewer for pointing this out and have added subsections in the “Material and Methods” section, including ‘Focal question’, ‘Search and information sources’ (this includes information on language and databases), ‘Eligibility criteria and study selection’, ’Outcomes assessed’, and ‘Quality assessment’.

  1. I recommend incorporating a quality assessment tool to evaluate the robustness of this systematic review. Choose from the various available tools and integrate it into your research to provide a comprehensive evaluation of the study's quality. In the "Materials and Methods" section, elucidate the methodology behind the selected quality assessment tool. Clearly outline the criteria, scoring system, and rationale behind its application, aiming to enhance transparency and credibility in the evaluation of your research findings. This inclusion will contribute to a more thorough understanding of the reliability and validity of your systematic review. Please integrate a table within the document to present a thorough quality assessment report, complemented by accompanying text for clarity and context.

We thank the reviewer for pointing this out. We regret that we have not made it clearer from the outset that the quality of the available studies was overall poor given that most were non-interventional open label extension studies and only 3 RCTs were found with results for naldemidine, the 2 naloxegol RCTs were unable to recruit sufficient patients. We used the CASP Randomized Controlled Trial Standard Checklist as outlined in the ‘quality assessment’ section under Methods, and added a ‘Limitations’ section under “Discussion’.

4. The discussion section of your manuscript requires improvement in its description. I recommend a thorough revision, where each included study is individually addressed within this section. Provide a comprehensive analysis of the main biases associated with each study, and discuss the primary and secondary endpoints to enhance the depth of your interpretation. Please ensure that the discussion does not duplicate information already presented in the table, aiming for a nuanced exploration of each study's nuances and contributing to a more insightful and well-rounded understanding of your research findings.

We have revised the discussion section of this manuscript completely.  Our table has a lot of detail and we added some more improvements to it as recommended by another reviewer. It would have become repetitive and beyond the scope to revisit each study individually in separate sections, thus we respectfully did not follow this reviewer’s suggestion in this respect. The added ‘limitation’ subsection sums up the biggest problems with each study reviewed.

5. In the "Authors" column of your table, kindly insert reference numbers corresponding to the reference list. This adjustment will ensure a more organized and systematic alignment between the authors' names and their respective references, facilitating easier cross-referencing for readers and enhancing the overall coherence of your presentation.

Thank you for this excellent suggestion to improve readability, the reference numbers have been added to the table, in addition to the authors’ name.

6. I request that you enhance and modify Figure 1 by incorporating the PRISMA chart, accessible at the provided link, to elucidate the search process employed in your manuscript. This adaptation will contribute to a clearer visualization of your study's systematic approach, aligning it with recognized standards and facilitating a more transparent representation of the search methodology for your readers.

We thank the reviewer for this excellent point and have revised Figure 1 to make it clearer.

7. Elevate the concluding section of your manuscript by paving the way for future research directions. Delve into the researchers' mindset, guiding them towards exploring novel studies that directly address the research findings, with an emphasis on benefiting future patients. Additionally, revisit your findings in greater depth, elucidating the clinical relevance of your results. By providing a thorough exploration of potential implications and applications, you can significantly enhance the impact and applicability of your research, contributing to the advancement of knowledge and its practical implications in the field.

We have added a ‘limitation’ section and reworded the ‘implication for pharmacy practice’ as well as the  conclusion section.

8. I believe that the inclusion of a structured abstract may not align seamlessly with the standards set by MDPI. It would be beneficial to reconsider the abstract format to ensure it adheres more closely to the specific guidelines and expectations established by MDPI for a more harmonious fit within their publishing standards.

We thank the reviewer for his comment.  The abstract was reworded and adheres to MDPI standards. The abstract does not appear in ‘track changes’ as it was no longer readable to us.

Reviewer 2 Report

Comments and Suggestions for Authors

The outcome of the study by Braun and coworkers is of clinical value because OIC is the most obstacle to continue treatment with opioid analgesics in patients with cancer pain.

My comments are as follows:

Abstract

Replace Peripherally-acting mu-receptor antagonists with Peripherally acting μ-opioid receptor antagonists (PAMORAs).

Abstract line 30 replace were with was. (Naldemedine or naloxegol was).

Introduction

Authors should provide the same nomenclature for μ-opioid receptors throughout the review.

Line 39-41: I suggest the authors to replace Opioids bring about analgesia by binding to μ-receptors with Opioid agonists bring about analgesia largely by binding to μ-opioid receptors (because Kappa agonists can also produce analgesia, but only few drugs are used in the management of pain).

Line 50-51 replace opiods’ with opioids’

I suggest the authors to cite the following paper ( https://doi.org/10.3390/molecules28237766) which is very close to the present work, besides 1-11 citations.

Structure: I suggest the authors to provide the chemical structures of the reviewed antagonists (Naloxegol and naldemedine) and the parent antagonists (naltrexone and naloxone).

Pharmacokinetics, Interactions and Contraindications: the written style of textbook was followed in these sections; scientific papers are great places to look for information. These sections lack references related to information provided by the authors. I ask the authors to refer to the sources of their data. Quotation number 20 is found only in the reference section but is not included in the main text.

Side effects: Authors should provide information related to the route of administration.

Table 2: I suggest the authors to give the author’s name and the number of the reference appeared in the text and reference section. Separate the therapeutic effect from the side effects in the result column (by space or dashed line). In the third and sixth studies the N of patients is missed (authors should verify).

Author Response

The outcome of the study by Braun and coworkers is of clinical value because OIC is the most obstacle to continue treatment with opioid analgesics in patients with cancer pain.

We thank the reviewers for their time in reviewing our manuscript and attention to detail. We have revised our manuscript taking all their excellent points into consideration. 

My comments are as follows:

Abstract

Replace Peripherally-acting mu-receptor antagonists with Peripherally acting μ-opioid receptor antagonists (PAMORAs).

Abstract line 30 replace were with was. (Naldemedine or naloxegol was).

Both was done, thank you.

Introduction

Authors should provide the same nomenclature for μ-opioid receptors throughout the review.

Line 39-41: I suggest the authors to replace Opioids bring about analgesia by binding to μ-receptors with Opioid agonists bring about analgesia largely by binding to μ-opioid receptors (because Kappa agonists can also produce analgesia, but only few drugs are used in the management of pain).

This was done.

Line 50-51 replace opiods’ with opioids’—Done.

I suggest the authors to cite the following paper ( https://doi.org/10.3390/molecules28237766) which is very close to the present work, besides 1-11 citations.

This citation was integrated into the paper, it is now reference #12.

Structure: I suggest the authors to provide the chemical structures of the reviewed antagonists (Naloxegol and naldemedine) and the parent antagonists (naltrexone and naloxone).

We have added the chemical structure of all 4 drugs.

Pharmacokinetics, Interactions and Contraindications: the written style of textbook was followed in these sections; scientific papers are great places to look for information. These sections lack references related to information provided by the authors. I ask the authors to refer to the sources of their data. Quotation number 20 is found only in the reference section but is not included in the main text.

We have expanded our references and added references to these sections, we also refer to the FDA and European Medicine Agency drug data on both medications.

Reference # 20 (now #24) is cited for the first time in the first paragraph of the ‘pharmacokinetics’ section of the paper, thank you for bringing this to our attention.

Side effects: Authors should provide information related to the route of administration.

We used the drugs’ FDA description RE side effects//route of administration.

Table 2: I suggest the authors to give the author’s name and the number of the reference appeared in the text and reference section. Separate the therapeutic effect from the side effects in the result column (by space or dashed line). In the third and sixth studies the N of patients is missed (authors should verify).

Thank you for this excellent suggestion to improve readability, the reference numbers have been added to the table, in addition to the authors’ name. We used a line to separate the therapeutic effect from the side effects in the results section of Table 2. We also added the N for the 2 studies to the table.

Reviewer 3 Report

Comments and Suggestions for Authors

I carefully read this review. This very interesting article is a synthesis of available data on naloxegol and naldemedine in 2024.

Good methodology and style.

The introduction should have a clinical focus on the alteration of the quality of life of patients, due to OIC. OIC can also alter analgesic strategies (like opioids change or doses decrease, with poor analgesia) and lead to a vicious circle with a big negative impact on patients' everyday life

I think PAMORAs should be presented as OIC-targeted therapies, which makes a difference from classic laxatives

line 153 please correct "pain,,"

I think the discussion and the conclusion should be enhanced, in particular discussing these points : 

- classic laxatives are ineffective in OIC, and they are responsible for side effects leading to their cessation, and poor compliance; classic counselings like hydration and physical activity, even if necessary in a global cancer context, are not effective on OIC but most of the time, they are the first messages to be delivered by pharmacist and physicians... this should be mentioned 

- please correct "Benfits Management Srvices " line 297

- the conclusion should enhance the role of pharmacists (especially clinical pharmacists) as major actors of supportive care in cancer by diagnosing OIC, counseling patients and physicians

Author Response

I carefully read this review. This very interesting article is a synthesis of available data on naloxegol and naldemedine in 2024.

We thank the reviewers for their time in reviewing our manuscript and attention to detail. We have revised our manuscript taking all their excellent points into consideration. 

Good methodology and style.

The introduction should have a clinical focus on the alteration of the quality of life of patients, due to OIC. OIC can also alter analgesic strategies (like opioids change or doses decrease, with poor analgesia) and lead to a vicious circle with a big negative impact on patients' everyday life

I think PAMORAs should be presented as OIC-targeted therapies, which makes a difference from classic laxatives

Both suggestions have been integrated, mainly in the second paragraph of the introduction but also later in the discussion.

line 153 please correct "pain,," – Done.

I think the discussion and the conclusion should be enhanced, in particular discussing these points : 

- classic laxatives are ineffective in OIC, and they are responsible for side effects leading to their cessation, and poor compliance; classic counselings like hydration and physical activity, even if necessary in a global cancer context, are not effective on OIC but most of the time, they are the first messages to be delivered by pharmacist and physicians... this should be mentioned 

We appreciate this excellent comment and have added this to the ‘implication for pharmacy practice’ section in the Discussion.

- please correct "Benfits Management Srvices " line 297 – Done,

- the conclusion should enhance the role of pharmacists (especially clinical pharmacists) as major actors of supportive care in cancer by diagnosing OIC, counseling patients and physicians

Thank you for the suggestion, we have rewritten the section on “implications for pharmacy practice’ as well as the conclusion.

Round 2

Reviewer 1 Report

Comments and Suggestions for Authors

Esteemed authors,

I extend my sincere gratitude for submitting the revised version of your manuscript. I am pleased to note the considerable improvements made in this iteration. After careful consideration, I wholeheartedly recommend the publication of this manuscript.